



# Case study of ozone anomalies over northern Russia in the 2015/2016 winter: Measurements and numerical modeling

Yury M. Timofeyev[1], Sergei P. Smyshlyaev[2], Yana A. Virolainen[1], Alexander S. Garkusha[1], Alexander V. Polyakov[1], Maxim A. Motsakov[2], Ole Kirner[3]

[1] Saint-Petersburg State University, 7/9, Universitetskaya Emb., St Petersburg, 199034, Russia

[2] Russian State Hydrometeorological University, 79 Voronezhskaya str., St. Petersburg, 192027, Russia

[3]Steinbuch Centre for Computing, Karlsruhe Institute of Technology, Kaiserstrasse 12, 76131 Karlsruhe Germany

*Correspondence to*: Sergei P. Smyshlyaev (smyshl@rshu.ru)

**Abstract.** Episodes of extremely low ozone columns were observed over the territory of Russia in the Arctic winter of

2015/2016 and the beginning of spring 2016. We compare total ozone columns (TOC) obtained using different remote sensing techniques (satellite and ground-based observations) and results of numerical modelling over the territory of the Urals and Siberia for the above period. We demonstrate that the provided monitoring systems (including new Russian Fourier- spectrometer IKFS-2) and modern 3-dimensional models are able to capture the observed TOC anomalies. However, the results of observations and modelling show discrepancies of up to 20-30% in TOC measurements. Analysis of

the role of chemical and dynamical processes demonstrates that it is unlikely that observed short-term TOC variability may be a result of local photochemical destruction initiated by heterogeneous halogen activation on particles of polar stratospheric clouds that formed under low temperatures in the mid-winter.

## 1 Introduction

Abnormally low values of total ozone columns (TOC) were recorded in January-February 2016 in the polar region of the

Northern Hemisphere (Zvyagintsev et al., 2016; Manney and Lawrence, 2016). Observed low values were recorded long before the beginning of spring, when chemical destruction of ozone occurs periodically in the Northern Hemisphere as a result of a strong vortex and the long existence of polar stratospheric clouds (PSCs) (Manney et al., 2011). Early anomalies in TOC indicate that it is unlikely that they may be caused by chemical disruption after the heterogeneous activation of chlorine and bromine gases on the surfaces of PSCs particles. The analysis of meteorological conditions during the

2015/2016 winter showed that during this period the lower polar stratosphere was extremely cold, which created a potential for a record ozone hole in the spring of 2016, but a strong sudden stratospheric warming in early March 2016 destroyed the polar vortex and prevented formation of a spring ozone anomaly (Manney and Lawrence, 2016). Nevertheless, during the entire winter of 2016 in the northern part of Russia, the ozone content was lower than in previous years, and the depth of

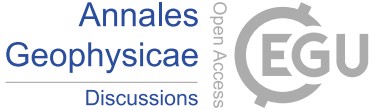

short-term ozone anomalies in January and February 2016 was comparable to the depth of the ozone mini-holes of the spring 2011.

Over recent decades, investigation of total ozone time-scale variations demonstrated regilar occurrence of the spring deep ozone depletion over the Antarctic region. This phenomenon was called the "ozone hole". In the Northern Hemisphere,

similar to the southern hemisphere polar column ozone depletion has been observed on smaller scale as well as over shorter time intervals, for example in 2011 (Manney et al., 2011; Balis, 2011). For episodes with extremely low TOCs (less than 220 Dobson units) these phenomena were called "ozone mini-holes" (Millan and Manney, 2017). Observation and prediction of the occurrence of episodes with abnormally low ozone content close to "mini-holes" is both crucial for the investigation of its nature and for the prediction of potential increase of UV-radiation on the Earth's surface. Unusually sharp and repetitive

TOC depletion was observed over the territory of the Urals and Siberia in the first quarter of 2016. In some cases, the TOC depletion reached 40-50% in comparison with climatic values (Zvyagintsev et al., 2016).

In this paper, we study the episodes of low TOCs over some Russian stations in January and February 2016 based on remote sensing observations and results of numerical modeling.

**2 Total ozone column measurements over Russia during winter 2016**

Monitoring of the total ozone level is provided by various ground-based remote sensing systems (Brewer and Dobson spectrophotometers, M-124 filter ozonometers, DOAS, Microwave and IR methods, Lidar measurements) and by various satellite systems (https://disc.gsfc.nasa.gov/; Timofeyev and Vasiliev, 2008; Staehelin et al., 2001). According to regular extensive validation programs (Balis et al., 2007; Boynard et al. 2016; Garkusha et al., 2017), total ozone measurement errors can be from 1–2 to 10% depending on the method, device, time and place of the measurements.

We analyzed the total ozone data of the first quarter of 2016, obtained by the basic Russian ground-based ozonometer M-124 and satellite instruments OMI and SBUV (recording outgoing solar reflected and scattered spectra of UV radiation), IASI and a new Russian instrument IKFS-2, recording outgoing atmospheric thermal IR radiation. The features of such satellite instruments as OMI, SBUV and IASI and the Russian ground-based ozonometer M-124 are well-known (Balis et al. 2007; Bhartia et al., 2013; Kroon et al., 2008; Viatte et al., 2011; Boynard et al., 2016). Independent assessments of TOC

measurement errors (Virolainen et al., 2017) showed values of 3.3–4.1 % for IASI, 2.0–3.5 % for M-124, and 1.9–2.1 % for OMI instruments. The infrared Fourier-transform spectrometer IKFS-2 on-board the satellite "Meteor-M" was launched in July 2014. IKFS-2 was preeminently designed for temperature-humidity sounding of the atmosphere and for measurement of some climatically important gases, including ozone. Detailed description of characteristics of IKFS-2 is given by Golovin et al. (2014). The advantage of the IKFS-2 and IASI instruments is its ability to conduct measurements in the absence of

sunlight, which is especially important for polar regions, where the polar night exists for a long time, during which the work of solar radiation measurement devices is impossible.

The description of the IKFS-2 measurement interpretation methodology, as well as estimates of errors in measurements of TOCs for cloudless and cloudy atmosphere, are given in the works (Garkusha et al., 2017; Garkusha et al., 2018). The





technique of interpretation, based on the method of artificial neural networks (ANN), is described in detail in the paper (Garkusha et al., 2017). The approximation of the solving operator of the inverse problem by a three-layer perceptron is used, the activation function of the neurons of the hidden layer is the hyperbolic tangent, the output neuron is linear. The main feature of the technique is the use as predictors of principle components (PC) the spectra measured by IKFS-2. The set

of predictors consists of 25 PC of the entire measured spectrum (660-2000 cm$^{-1}$), 50 PC only of the ozone absorption band and the measurement zenith angle. For ANN training, the results of TOC measurements using the OMI instrument from the AURA satellite were used (McPeters et al, 2015). Estimates of the error in determining the TOCs with IKFS-2 are on the average in the range 2-6%. The largest differences (up to 10%) are observed in the southern polar latitudes in the presence of an ozone hole over Antarctica.

In the first quarter of 2016, three long periods of essential deviations of the average daily TOCs from average long-term values were registered over the territory of Russia. TOCs decreases reached: 39–52% (in 26.01–01.02 over the Northern regions of the Urals and Siberia), 30–50% (in 20.02–03.03 over Northern Siberia), 27-39% (in 09–19.03 over Central Siberia) of average values (191–257, 227–321, 257–321 DU, respectively) (Zvyagintsev et al., 2016). Extremely low winter TOC values (episodically less than mini-hole threshold) were observed over the northern regions of the Urals and Siberia for

the first time. During January 27–31 TOCs smaller than 220 DU were recorded at Russian ozonometric network stations using M-124 measurements (Pechora, 65°N, 57°E; Khanty-Mansiysk, 61°N, 69°E; Turukhansk, 66°N, 88°E; Round, 64°N, 100°E) and by OMI devices on the board of Aura satellite.

Figure 1 taken from (Garkusha et al 2018) depicts the spatial distribution of TOCs in February 23-27, 2016, based on measurements of two instruments of the same type - IKFS-2 and IASI. The figure shows good agreement between two

independent satellite measurements.

Figure 2 presents the evolution of TOCs measured at three ground-based observational stations: Tura (61° N, 100° E), Pechora, and Khanty-Mansyisk. The comparison allows us to draw the following conclusions:

(a)    All instruments and measurement methods generally provide a good description of the main features of TOC time variations, including observed ozone depletion. For the whole period of comparison, the average differences in the

results obtained by different types of measurements in most cases are 1–5%, with standard deviations of 3–8%.

(b)    The only exception is the IASI measurements in Khanty-Mansiysk and Tura, for which the standard deviations of the differences with ground-based measurements in the first quarter of 2016 reach 12%. In addition, at the Tura station, IASI data overstates the M-124 measurements by 11% on average.

(c)    All satellite data overestimate the values of TOC in comparison with ground-based measurements during the

periods of ozone depletion. This is, possibly, due to the fact that the optimal retrieved solution is constructed not only from atmospheric radiation spectra that have been measured, but it also employs *a priori* information about TOCs. This then described the mean state of ozone concentrations and not a mini-hole event. To exclude this effect, it is necessary to improve the *a priori* information by incorporating the ozone total column amount with the appearance of strong ozone anomalies.



### 3 Comparison of total ozone column measurements and numerical modeling

Also relevant to this issue is the comparison of observational data with the results of numerical modeling. Values of TOC were calculated by two three-dimensional atmospheric chemistry models, which take into account observed variations of meteorological parameters based on re-analysis of the measurement results, the chemistry-climate model (CCM) EMAC (ECHAM/MESSy Atmospheric Chemistry model) (Jöckel et al., 2006) and the Russian State Hydrometeorological university chemistry-transport model (RSHU CTM) (Smyshlyaev et al., 2017).

The EMAC model is a numerical chemistry and climate simulation system that includes tropospheric and middle atmosphere processes (Jöckel et al., 2010). It uses the second version of the Modular Earth Submodel System (MESSy2). The core atmospheric model is the 5th generation European Centre Hamburg general circulation model (ECHAM5, Roeckner et al. 2006). The core model, ECHAM5, uses a spectral transform technique, the so-called T-value indicating the degree of triangular spectral truncation. For the present study, we applied EMAC (ECHAM5 version 5.3.02, MESSy version 2.52) in T42 resolution; i.e., with a spherical truncation of T42 (corresponding to a quadratic grid of 2.8 x 2.8 degrees, respectively, in latitude and longitude. Vertically, the model resolves the troposphere, stratosphere and lower mesosphere (39 hybrid levels from the surface up to 0.01 hPa, about 80 km). We applied a Newtonian relaxation technique (Nudging) to our model simulation with the help of the ERA-INTERIM reanalysis data set (Dee et al., 2011) to improve consistence between the simulated and observed temperature and wind fields responsible for the dynamical impact on ozone distribution. A detailed description of the EMAC model and its applications can be found in (Righi et al., 2015, Virolainen et al., 2016).

The global RSHU CTM is based on the Institute of Numerical Mathematics and Russian State Hydrometeorological University (INM RAS – RSHU) CCM (Galin et al., 2007), but meteorological fields are not calculated but specified from the Modern-Era Retrospective Analysis for Research and Applications (MERRA) reanalysis (Rienecker et al., 2011). The model has 5 x 4 degrees horizontal resolution in longitude by latitude and 31 vertical sigma levels from the surface up to approximately 60 km. The distribution of the oxygen, hydrogen, nitrogen, chlorine, bromine and carbon gases are calculated in the manner described by Smyshlyaev et al. (1998). PSCs formation and evolution is taken into account according to Smyshlyaev et al. (2010).

The analysis of comparison between modeling and experimental (OMI, version TOMS) leads to the following conclusions (Fig.3):

- Both models sufficiently describe time variations of the total ozone content. On average, the RSHU model provides 1–2% smaller values of the TOC than those observed by OMI. EMAC, conversely, exceeds the OMI measurements by 7–9%. The standard deviations for both models are 6–7%. This approaches the standard deviations between different types of measurements of the total ozone content during the examined period.

- EMAC better describes the TOC variations during some depletion periods than the RSHU CTM model: at the Khanty-Mansiysk station standard deviation stood at 4–5% for EMAC model whereas the RSHU model ranged between 6 and 8%. At the Tura station during the January minima, on the contrary, the RSHU model is in better agreement with OMI





measurements (3% vs. 7%). Neither model describes the observed January mini-holes at the Pechora station (standard deviations reach 12–15%).

- On certain days, the differences between measurements and modeling can be up to 20–30%. Models often overestimate the total ozone content measured by the OMI instrument (especially the EMAC model).

**4. Analysis of the processes that define observed ozone variability over Russia during the Arctic winter of 2015/2016**

The role of chemical and dynamic processes in the observed TOC variability over Russia was assessed based on the RSHU CTM calculations. Two days with the lowermost TOCs registered at all stations were selected for extended analysis. These days are January 27, 2016 (day 27) and February 19, 2016 (day 50) (Fig.3). Results of RSHU CTM simulations for these days are presented in Fig.4 for column ozone (top figures) together with the MERRA temperature data averaged for the lower stratosphere (14–25 km) (bottom figures). The regions with low TOCs are consistent with the low stratospheric temperatures. This is a result of dynamical isolation, which leads to stratospheric cooling and ozone depletion as a result of heterogeneous chemical reactions on PSCs particles leading to chlorine activation.

The surface area of the PSCs for the days with low stratospheric temperature and low column ozone episodes are presented in Fig. 5 (top panel). Enhanced PSCs surface area is located at the same regions where low stratospheric temperatures were registered. This is obvious consequence of stratospheric cooling and may lead to heterogeneous chlorine and bromine activation followed by ozone depletion similar to the Antarctic ozone hole formation (Solomon, 1999). In order to evaluate local ozone destruction significance for the observed TOC depletion, the photochemical ozone loss coefficient ($s^-$ $^1$) (Jacobson, 2005) (rate of ozone loss divided by the ozone concentration: $\Lambda_{O3} = L_{O3} / N_{O3}$, where $L_{O3}$ - is photochemical ozone loss (mol/s/cm$^3$) and $N_{O3}$ is ozone concentration (mol/cm$^3$)), calculated with the RSHU CTM, is presented in the bottom panel of Fig.5.

The location of zones with enhanced ozone destruction is close to the regions with estimated low TOCs, but is not fully consistent. In addition, the minimum local photochemical ozone lifetime, estimated as a reciprocal of the ozone destruction coefficient, is about 200 days under these days' conditions. Such a long photochemical lifetime of ozone may be treated as a sign of the unlikeliness that the observed short-term ozone variability may be a result of local photochemical destruction initiated by heterogeneous chlorine and bromine activation on the particles of PSCs that formed in these regions. On the other hand, simultaneous low stratospheric cooling and low column ozone at the same locations may be caused by dynamic divergence that leads to heat and mass deficit, similar to polar vortex isolation (Solomon, 1999). Another confirmation of the prevalent dynamical nature of the observed episodes with low ozone concentration is their formation during polar night recorded during December 2015 and first part of January 2016 when photochemical destruction is negligible.



### 5. Summary

Data analysis and numerical model experiments have been used to analyze the low TOCs recorded over Russia during the 2015/2016 winter. Ozone anomalies were observed over the territory of the Urals and Siberia in winter and the beginning of spring 2016. In this paper, we compare TOCs obtained using different measuring methods (satellite and ground-based

observations) and results of numerical modeling (for the stations Khanty-Mansiysk, Tura, and Pechora) for the above period. It is shown that existed monitoring systems (including Russian Fourier- spectrometer IKFS-2) and modern 3-dimensional models provide a good description of the occurrence of the TOC anomalies. However, results of observations and modeling diverge on particular days by as much as 20-30%. Analysis of the role of chemical and dynamical processes in the observed ozone variability over the Russian Federation was based on the RSHU CTM calculations. This analysis demonstrated that it

is unlikely that local photochemical destruction initiated by heterogeneous halogen activation on the particles of PSCs that formed under registered low temperatures may be responsible for short-term local ozone destruction. The prevalent reason for the observed low TOCs may be dynamical flux divergence out of regions with observed low ozone content (Smyshlyaev et al., 2017).

### Acknowledgments

The comparison and analysis of different experimental and simulated total ozone column data was supported by the Russian Science Foundation (Grant #14-17-00096). The modeling of Arctic processes with the RSHU CTM was supported by the Russian Foundation for Basic Research (Grant #17-05-01277). The interpretation of Russian Fourier- spectrometer IKFS-2 spectra was supported by the Russian Foundation for Basic Research (Grant #17-05-00768). The discussions of the results and the retrieval of EMAC model data was supported by Saint Petersburg University (Grant #11.42.690.2017). The

authors thank A. M. Zvyagintsev from the Central Aerological Observatory (Dolgoprudny, Russia) for providing the ozone measurements from the ground-based network. Both the simulation data and measurement results necessary to reproduce the comparison are available from the authors upon request (yana.virolainen@spbu.ru).

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





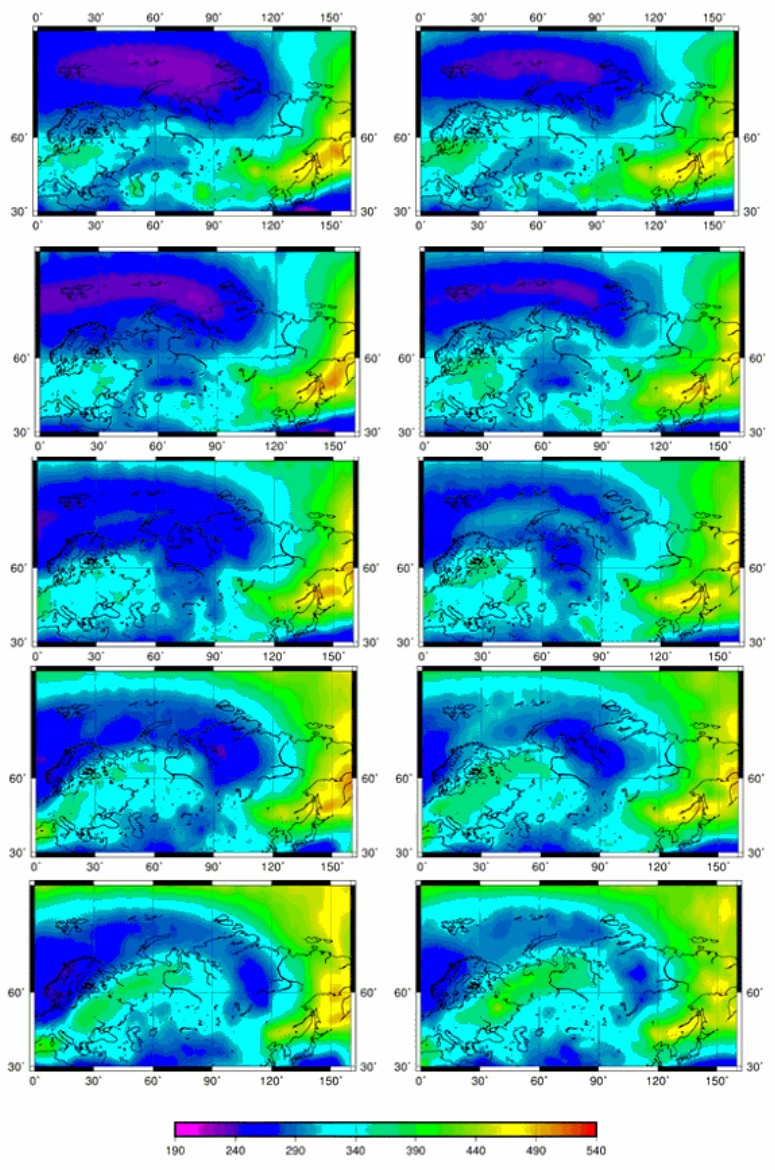

**Figure 1. Spatial distributions of the total ozone columns in February 23 - 27, 2016, based on measurements of two instruments of the same type - IKFS-2 (left) and IASI (right). (Garkusha et al, 2018).**



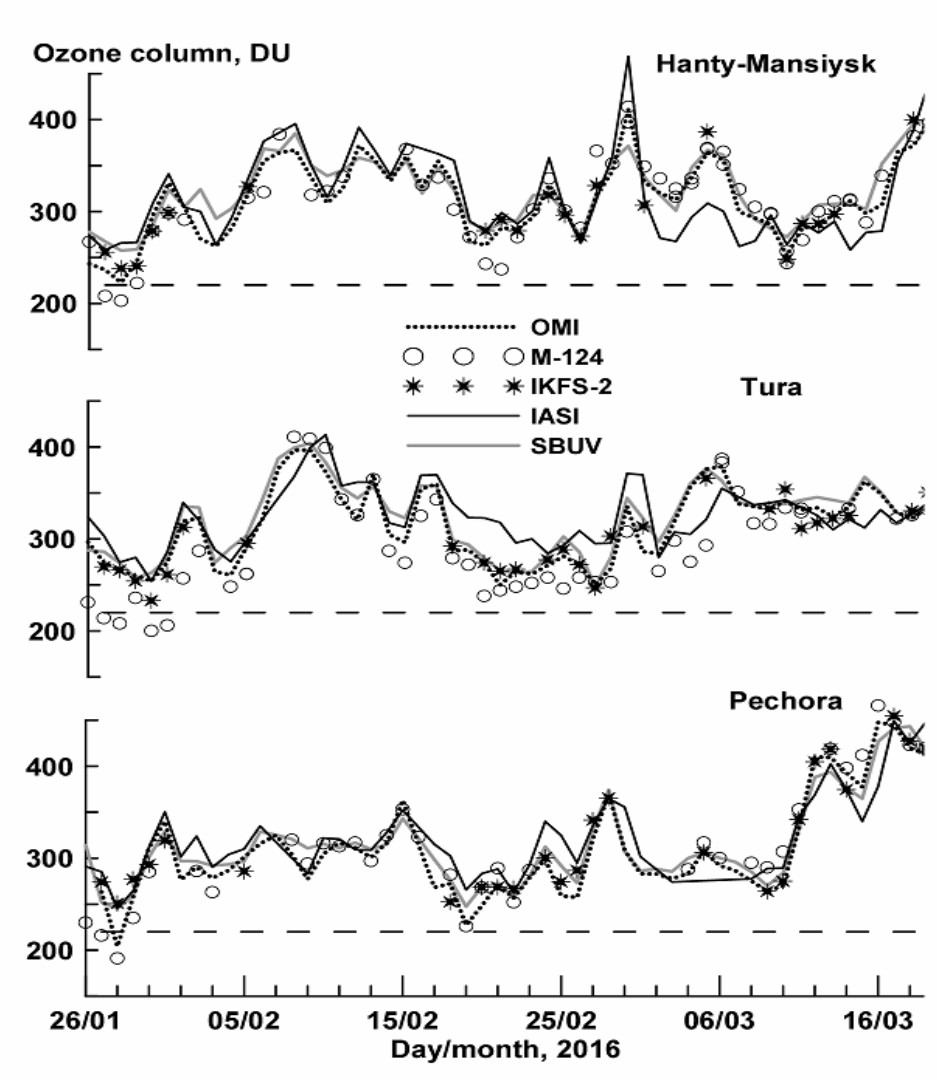

**Figure 2:** Total ozone measurements provided by the OMI, M-124, IKFS-2, IASI, and SBUV for Khanty-Mansiysk, Tura, and Pechora stations. A dotted line indicates the threshold value of total ozone content, defined as ozone mini-holes.



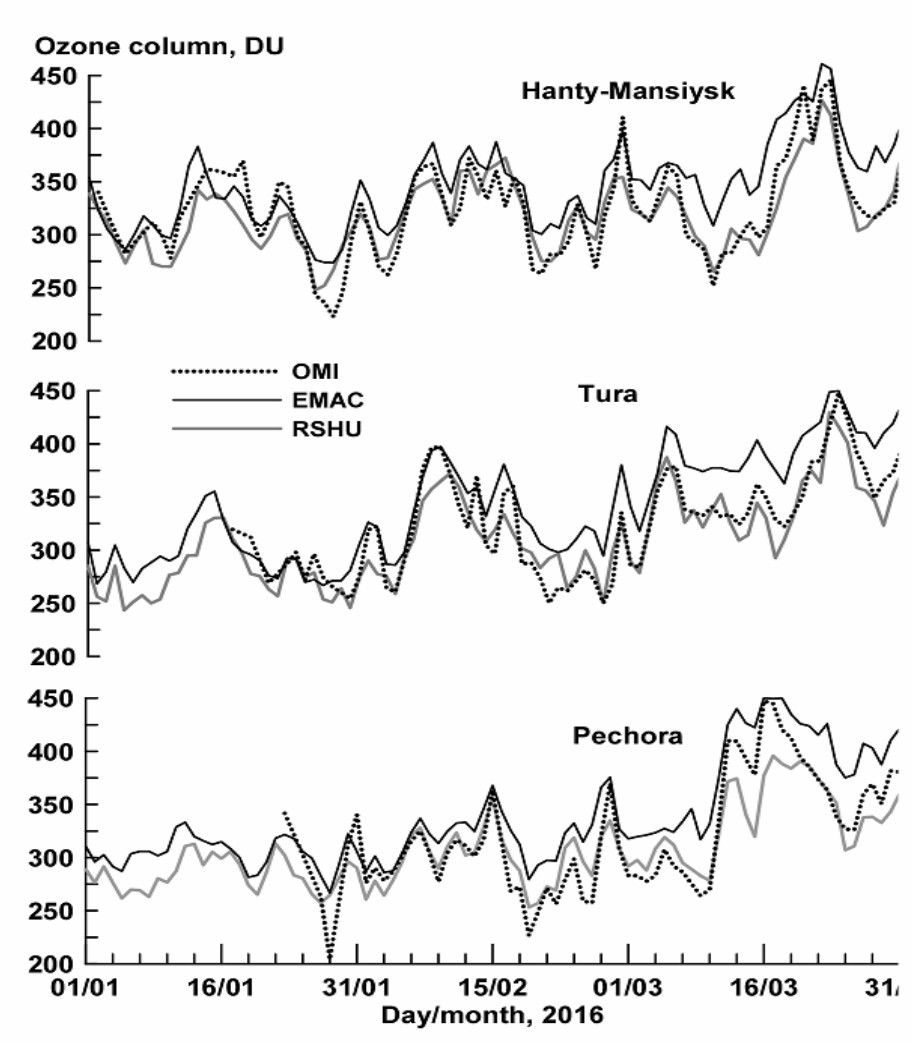

**Figure 3:** Total ozone measurements provided by OMI and modeling results from EMAC and RSHU for the stations Khanty-Mansiysk, Tura, and Pechora.



**Figure 4: Column ozone (Dobson units) for days with minimum local registered values (*top panels*) and temperature of the lower stratosphere (K) for the same days (*bottom panels*) simulated with the RSHU model.**

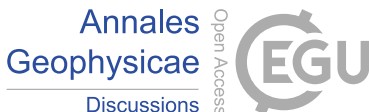





**Figure 5: Calculated with RSHU CTM low stratospheric polar stratospheric clouds surface area (10$^8$ cm$^2$/cm$^3$) for days with minimum local registered column ozone values (*top panel*) and averaged for the low stratosphere ozone loss coefficient (10$^8$s$^{-1}$) for the same days (*bottom panel*).**