# Peer review of "Case study of ozone anomalies over northern Russia in the 2015/2016 winter: Measurements and numerical modelling"

_Annales Geophysicae, 2018_

## Referee Comment (RC1) · Anonymous Referee #1 · 3 Apr 2018

Review of the paper "Case study of ozone anomalies over northern Russia in the 2015/2016 winter: Measurements and numerical modeling" by Y. M. Timofeyev,et al.

General comments

The paper discusses a case study of ozone anomalies over N. Russia and assesses the role of chemical and dynamical processes in observed short-term ozone loss, using observations as well as CCM and CTM simulations. The paper contains new material.

However, improvements are needed before accepted for publication.

Overall, a clearer description is needed not only on the methodology but also on its

application and the interpretation of results, as some paragraphs may be even confusing. For example, Section 3 on the "Comparison of total ozone column measurements and numerical modeling", one of the papers' essential sections, does not discuss at all the rationale of the use of two different models. Moreover, the two different models are forced (or nudged) with different data sets. Both of these facts introduce errors and discrepancies in the comparisons, which are not discussed at all. Moreover, there is no information on the comparison between the two models and the comparison between the two forcing data sets. Please correct also the figures, and clarify the Figure Captions. Figure 2 and Figure 3 contain solid lines with different shades of gray, and it can be hard to distinguish between them.

Specific comments:

1. Page 2, lines 10-11 end elsewhere in the paper: "TOC depletion..." Short term episodes of low ozone values are better described by "ozone loss" rather than depletion, a term which implies a longer-term decay with significant duration

2. Page 3, line 10: "three long periods of essential deviations of the average daily TOC..." I do not understand what is meant here. Please clarify

3. Page 3, line 10:"... from average long-term values..." What is the time period for this climatological average that you refer to?

4. Page 3, line 13:"of average values (191-257,...)" Please clarify what is this range of values referring to?

5. Page 3, line 18. Please indicate the location of the stations in Figure 1, and/or in Figures 4 and 5. It will be extremely helpful for the reader's understanding.

6. Page 3, line 24 "including observed ozone depletion". Do you mean here the ozone depletion in general (e.g. Northern hemisphere), or short-term ozone loss? Please clarify.

7. Page 3, line 30: "..during periods of ozone depletion" Again, does this refer to a

general longer term behavior, or the short period examined here?

8. Page 3, line 31-32:" This then described the mean state . . ." I do not understand the meaning of this sentence

9. Also in line 32-33, "to improve. . . with the appearance of strong ozone anomalies". Do you mean inclusion of ozone anomalies? It is not clear at all.

10. Page 4, "Comparison of total ozone column . . ." Section Please see my general comment in the beginning

11. .. Page 4, line 25: Figure 3. It is not clear which line is for each model. I can see (and print) two gray lines. Please correct.(also in Fig. 2)

12. Page 5, lines 10 and below: What is printed in the lower panels of Figure 5? The text refers to MERRA, but the figure caption refers to model output.

---

## Referee Comment (RC2) · Anonymous Referee #2 · 26 Apr 2018

The manuscript is aimed at the study of low total column ozone (TCO) episodes over the northern Russia during the winter 2015/2016. The authors analyzed the ground based and satellite TCO observations and compare them with the output of two models (RSHU and EMAC) exploited in specified dynamics modes. The subject of the manuscript is appropriate for AnGeo. The manuscript is well written and structured, the figures and explanations are clear. Despite of many similar studies have been described in the literature the manuscript provides some new information about the available ozone observing systems. The analysis of the observed data is extended by the comparison with numerical simulations and attempt to attribute the causes behind the appearance of low TCO events. There are, however, some issues which should be

clarified before the publication of the manuscript.

Major issue

The conclusion about the role played by chemical and transport processes is not strongly supported by model results. It is mostly based on general knowledge of the different ozone time scales and not strongly supported by model results. The authors show that the RSHU model simulates some enhancement of PSC occurrence and ozone loss rate in the low TCO regions, but do not use it to support or reject the importance of chemical ozone loss. I think it makes sense to run specially designed model experiment (say the run without heterogeneous chemistry) to support purely dynamical nature of the low TCO events or explain why such runs cannot be performed.

Minor issues: 1. Page 1, Line 15: "unlikely" sounds too weak for the abstract.

2. Page 1, Lines 23-24: I would avoid using the same sentences in the introduction and conclusions. Potential readers could wonder if the minor role of heterogeneous chemistry is well known than why to tackle this issue again?

3. Page 1, Line 28: 2015/2016

4. Introduction: The motivation for the presented study should be stronger. The authors should emphasize the necessity to analyze new instruments and exploit two different models.

5. Ozone depletion: I understand this as chemical processes. However, low TCO events can be explained by the transport of low ozone to the considered location from the area inside polar vortex, where the ozone is small because of suppressed influx from the ozone production area. The authors concluded that the contribution of chemical destruction is small. Maybe than the ozone depletion term is not perfectly correct?

6. PSC: The ozone depletion via heterogeneous chemistry strongly depends on the availability of liquid sulfate aerosols. How they are treated in the models? The authors show PSC area. Does it include all kind of PSCs or just NAT?

7. Section 3: The exploited models use different meteorological reanalysis, therefore the difference in the results can be attributed to either model or reanalysis features. Would it be possible to attribute more precisely the difference between model results?

8. Page 5, lines 11-13: I am not completely agree with this statement. Does it mean that low TCO inside polar vortices will not take place without heterogeneous chemistry and chlorine activation. I think the role of transport is more important.

9. Page 5, last paragraph: In the present form it is not instructive. See my major comment.

10. Figures 4,5: The numbers are too small and hardly visible.

11. Figure 5: My impression is that the ozone loss about 10(8) molecules per second is too large. Please, check.

---

## Author Response (AR1)

**Author's response**

**Case study of ozone anomalies over northern Russia in the 2015/2016 winter: Measurements and numerical modeling**

5 By Yury M. Timofeyev, Sergei P. Smyshlyaev, Yana A. Virolainen, Alexander S. Garkusha, Alexander V. Polyakov, Maxim A. Motsakov, Ole Kirner

Abstract. Episodes of extremely low ozone columns were observed over the territory of Russia in the Arctic winter of 2015/2016 and the beginning of spring 2016. We compare total ozone column (TOC) obtained using different remote sensing techniques (satellite and ground-based observations) and results of numerical modelling over the territory of the

- 10 Urals and Siberia for the above period. We demonstrate that the provided monitoring systems (including new Russian Fourier- spectrometer IKFS-2) and modern 3-dimensional models are able to capture the observed TOC anomalies. However, the results of observations and modelling show discrepancies of up to 20-30% in TOC measurements. Analysis of the role of chemical and dynamical processes demonstrates that observed short-term TOC variability is not a result of local photochemical loss initiated by heterogeneous halogen activation on particles of polar stratospheric clouds that formed under
- 15 low temperatures in the mid-winter.

**Reply to reviewer 1**

**Dear Referee,**

**20**

Thank you for your comments on the paper and constructive recommendations. We have tried to follow your suggestions and have utilized most of them. Following we mention how the manuscript has been changed according to your comments.

**25 General comments:**

For example, Section 3 on the "Comparison of total ozone column measurements and numerical modeling", one of the papers' essential sections, does not discuss at all the rationale of the use of two different models. Moreover, the two different models are forced (or nudged) with different data sets. Both of these facts introduce errors and

discrepancies in the comparisons, which are not discussed at all. Moreover, there is no information on the comparison between the two models and the comparison between the two forcing data sets. Please correct also the figures, and clarify the Figure Captions. Figure 2 and Figure 3 contain solid lines with different shades of gray, and it can be hard to distinguish between them.

5

**ACCEPTED**

- 1. In Section 3, the motivation for using two different models is added, consisting in an attempt to assess the impact of the interactive interaction of the chemical and dynamical processes (the
- 10 EMAC model) with a re-analysis data nudging against the background of using re-analysis data in the RSHU model. In addition, the models have different spatial resolution, which makes it possible to estimate the effect of model resolution on the comparison with the observations related to the local points.
  - 2. An additional numerical experiment using the ERA-INTERIM reanalysis data was performed
  - with the RSHU model in order to compare the effect of different meteorological data on the comparison of the results of numerical modeling and local observations.
    - 3. At the end of section 3, the results of a comparison of numerical modeling and observations, as well as comparisons between models with different meteorological data, are expanded.
    - 4. Figures 2 and 3 are made in color, and the RSHU model simulation results with the EPA-
- 20

15

INTERIM data are added in Figure 3.

**Specific comments.**

1. Page 2, lines 10-11 end elsewhere in the paper: "TOC depletion: : :" Short term episodes of low ozone values are better described by "ozone loss" rather than depletion, a term which implies a longer-term decay with

25 significant duration

2. Page 3, line 10: "three long periods of essential deviations of the average daily TOC: : :" I do not understand what is meant here. Please clarify

3. Page 3, line 10:": : : from average long-term values: : :" What is the time period for this climatological average that you refer to?

4. Page 3, line 13:"of average values (191-257,:::)" Please clarify what is this range of values referring to?
5. Page 3, line 18. Please indicate the location of the stations in Figure 1, and/or in Figures 4 and 5. It will be extremely helpful for the reader's understanding.

6. Page 3, line 24 "including observed ozone depletion". Do you mean here the ozone depletion in general (e.g. Northern hemisphere), or short-term ozone loss? Please clarify.

35 7. Page 3, line 30: "...during periods of ozone depletion" Again, does this refer to a general longer term behavior, or the short period examined here?

8. Page 3, line 31-32:" This then described the mean state : : :" I do not understand the meaning of this sentence 9. Also in line 32-33, "to improve: : : with the appearance of strong ozone anomalies". Do you mean inclusion of ozone anomalies? It is not clear at all.

10. Page 4, "Comparison of total ozone column : : :" Section Please see my general comment in the beginning

5 11. .. Page 4, line 25: Figure 3. It is not clear which line is for each model. I can see (and print) two gray lines. Please correct.(also in Fig. 2)

12. Page 5, lines 10 and below: What is printed in the lower panels of Figure 5? The text refers to MERRA, but the figure caption refers to model output.

**10 ACCEPTED**

1. Page 2, lines 10-11 and elsewhere. The depletion of ozone is everywhere replaced by ozone loss.

2. Page 3, line 10-14. Clarification has been done.

**15**

3. Page 3, line 10. Climatological period is specified to be from 1979 to 2017.

4. Page 3, line 13. Clarification for the ranges of values has been done.

- 5. Page 3, line 18. Location of stations has been indicated at the fig.2.
  - 6. Page 3, line 24. Clarification for short-term ozone loss have been done.
  - 7. Page 3, line 30. Short-term period of ozone loss is clarified.

**25**

- 8. Page 3, line 31-32. The sentence has been modified.
- 9. Page 3, line 32-33. The sentence has been corrected.
- 30 10. Page 4. Section 3 has been extended with a more detailed comparison for observations and modeling.
  - 11. Page 4, line 25. Figures 2 and 3 now are plotted in color.

- 12. Page 5, lines 10 and below. Figure 4 captures are corrected top panels are for model output, and low panels for MERRA data.
- 5 Thank you again for taking the time to review our manuscript.

With respect,

Yu.M.Timofeyev, S.P.Smyshlyaev, Ya.A.Virolainen, A.S.Garkusha, A.V.Polyakov, M.A. Motsakov, O.Kirner.

10

**Reply to reviewer 2**

Dear Referee,

5 Thank you for your comments on the paper and constructive recommendations. We have tried to follow your suggestions and have utilized most of them. Following we mention how the manuscript has been changed according to your comments.

**Major issue:**

- 10 The conclusion about the role played by chemical and transport processes is not strongly supported by model results. It is mostly based on general knowledge of the different ozone time scales and not strongly supported by model results. The authors show that the RSHU model simulates some enhancement of PSC occurrence and ozone loss rate in the low TCO regions, but do not use it to support or reject the importance of chemical ozone loss. I think it makes sense to run specially designed model experiment (say the run without heterogeneous chemistry) to support purely dynamical nature of the low
- 15 TCO events or explain why such runs cannot be performed.

**ACCEPTED**

Two additional numerical experiments were carried out with RSHU CTM to confirm the conclusions

- 20 about the dominant role of the dynamical processes in the observed short-term ozone loss: one did not take into account the formation of polar stratospheric clouds in the Arctic zone, and the second did not take into account the chemical destruction of ozone to the north of the northern polar circle. A comparison of the three model experiments for the three stations considered in this paper is shown in Fig. 6. The results of model experiments have shown that the main features of the short-term ozone loss
- 25 are reproduced even without taking into account chemical destruction within the polar zone. At the same time, the influence of chemical processes becomes noticeable at the end of March, especially for Pechora.

Minor issues.

1. Page 1, Line 15: "unlikely" sounds too weak for the abstract.
2. Page 1, Lines 23-24: I would avoid using the same sentences in the introduction and conclusions. Potential readers could wonder if the minor role of heterogeneous chemistry is well known than why to tackle this issue again?

3. Page 1, Line 28: 2015/2016

4. Introduction: The motivation for the presented study should be stronger. The authors should emphasize the necessity to analyze new instruments and exploit two different models.

5. Ozone depletion: I understand this as chemical processes. However, low TCO events can be explained by the

5 transport of low ozone to the considered location from the area inside polar vortex, where the ozone is small because of suppressed influx from the ozone production area. The authors concluded that the contribution of chemical destruction is small. Maybe than the ozone depletion term is not perfectly correct?

6. PSC: The ozone depletion via heterogeneous chemistry strongly depends on the availability of liquid sulfate aerosols. How they are treated in the models? The authors show PSC area. Does it include all kind of PSCs or just NAT?

7. Section 3: The exploited models use different meteorological reanalysis, therefore the difference in the results can be attributed to either model or reanalysis features. Would it be possible to attribute more precisely the difference between model results?

8. Page 5, lines 11-13: I am not completely agree with this statement. Does it mean that low TCO inside polar

15 vortices will not take place without heterogeneous chemistry and chlorine activation. I think the role of transport is more important.

9. Page 5, last paragraph: In the present form it is not instructive. See my major comment.

10. Figures 4,5: The numbers are too small and hardly visible.

11. Figure 5: My impression is that the ozone loss about 10(8) molecules per second is too large. Please, check.

20

10
1. Page 1, line 15: The sentence corrected with replacement "unlikely" to more strong statement. 25

2. Page 1, lines 23-24. The sentence modified with attention shifted from the statement to a question whether chemical destruction on the surface of polar stratospheric clouds, for which a long existence of PSCs is necessary, to be responsible for the observed anomalies, or other factors, especially dynamic ones, would have a greater effect on the observed features.

**30**

3. Page 1, line 28. Corrected form 2016 to 2015/2016.

4. Introduction. The motivation extended with a justification of the need for additional measurements and numerical model experiments.

**35**

5. The term "ozone depletion" almost everywhere in the paper is replaced by "ozone loss".

6. A short description of the PSC formation and evolution code with appropriate references is added to the section 3. The code accounts for STS, NAT and ICE particles formation on the base of sulfur40 aerosol.

7. The motivation for using two different models is added. An additional numerical experiment using the ERA-INTERIM reanalysis data was performed with the RSHU model in order to compare the effect of different meteorological data on the comparison of the results of numerical modeling and local observations. The results of a comparison of numerical modeling and observations, as well as comparisons between models with different meteorological data, are expanded.

8. Page 5, lines 11-13. The sentence has been modified with a shift to a chance of heterogeneous ozone destruction which is checked at the following discussion. "This is a result of dynamical isolation, which leads to stratospheric cooling and potentially may cause ozone depletion as a result of heterogeneous chemical reactions on PSCs particles leading to chlorine activation."

10 9. Page 5, last paragraph. Results of additional numerical experiments without PSC processing included are added into discussion to demonstrate the prevalent role of dynamical processes in the observed short-term ozone loss.

10. Figures 4 and 5 are corrected to make numbers more visible.

11. Figure 5. Corrected to 10(-8). Actually this means that ozone loss coefficient was multiplied by

15 10(8) before plotting.

5

Thank you again for taking the time to review our manuscript.

With respect,

Yu.M.Timofeyev, S.P.Smyshlyaev, Ya.A.Virolainen, A.S.Garkusha, A.V.Polyakov, M.A. Motsakov, O.Kirner.

**20 REVISED PAPER WITH CHANGES MARKED**

**Case study of ozone anomalies over northern Russia in the 2015/2016 winter: Measurements and numerical modeling**

Yury M. Timofeyev1, Sergei P. Smyshlyaev2, Yana A. Virolainen1, Alexander S. Garkusha1, Alexander V. Polyakov1, Maxim A. Motsakov2, Ole Kirner3

[revised manuscript text omitted]

---

## Author Response (AR2)

**Author's response**

**Case study of ozone anomalies over northern Russia in the 2015/2016 winter: Measurements and numerical modeling**

5  By Yury M. Timofeyev, Sergei P. Smyshlyaev, Yana A. Virolainen, Alexander S. Garkusha, Alexander V. Polyakov, Maxim A. Motsakov, Ole Kirner

**Abstract.** Episodes of extremely low ozone columns were observed over the territory of Russia in the Arctic winter of 2015/2016 and the beginning of spring 2016. We compare total ozone columns (TOC) from different remote sensing techniques (satellite and ground-based observations) with results of numerical modelling over the territory of the Urals and

10  Siberia for this period. We demonstrate that the provided monitoring systems (including the new Russian Infrared Fourier Spectrometer IKFS-2) and modern 3-dimensional atmospheric models can capture the observed TOC anomalies. However, the results of observations and modelling show differences of up to 20-30% in TOC measurements. Analysis of the role of chemical and dynamical processes demonstrates that the observed short-term TOC variability is not a result of local photochemical loss initiated by heterogeneous halogen activation on particles of polar stratospheric clouds that formed under

15  low temperatures in the mid-winter.

**Reply to Topical Editor**

Dear Topical Editor,

20  Thank you for your comments on the paper and constructive recommendations. We have carefully checked the manuscript and fixed all typos found with an assistance of the English native speaker. Following we mention how the manuscript has been changed after corrections.

Thank you again for taking the time to review our manuscript.

25  With respect,

Yu.M.Timofeyev, S.P.Smyshlyaev, Ya.A.Virolainen, A.S.Garkusha, A.V.Polyakov, M.A. Motsakov, O.Kirner.

**Reply to reviewer 1**

Dear Referee,

Thank you for your comments on the paper and constructive recommendations. We have thoroughly checked the manuscript and tried to fix the potential typos and clarify the meanings. In particular, you are absolutely right regarding the sentence at section 3, page 4, lines 10-11. We tried to assess the differences between the CCM response and the CTM response. At the revised version we corrected this sentence. At the end of section 3, page 5, lines 20-24 we presented conclusion about this comparison.

"In general, the comparison of the EMAC and RSHU simulation, which both use the ERA-Interim reanalysis, and OMI, demonstrate that as well the interactive coupling between dynamical and chemical processes, as the different spatial resolutions do not have a principal influence on the quality of the representation of the short-term column ozone variability at local points. Both models show a good qualitative agreement with the OMI satellite observations, while for some local points and time periods the best agreement is shown by the EMAC CCM, and for others by the RSHU CTM".

Following we mention how the manuscript has been changed.

Thank you again for taking the time to review our manuscript.

With respect,

Yu.M.Timofeyev, S.P.Smyshlyaev, Ya.A.Virolainen, A.S.Garkusha, A.V.Polyakov, M.A. Motsakov, O.Kirner.

**Reply to reviewer 2**

Dear Referee,

Thank you for taking the time to review our manuscript and your positive decision.

With respect,

Yu.M.Timofeyev, S.P.Smyshlyaev, Ya.A.Virolainen, A.S.Garkusha, A.V.Polyakov, M.A. Motsakov, O.Kirner.

**REVISED PAPER WITH CHANGES MARKED**

[revised manuscript text omitted]